# AdipoRon, adiponectin receptor agonist, improves vascular function in the mesenteric arteries of type 2 diabetic mice

**Soo-Kyoung Choi, Youngin Kwon, Seonhee Byeon, Chae Eun Haam, Young-Ho Lee***

Department of Physiology, College of Medicine, Brain Korea 21 PLUS Project for Medical Science, Yonsei University, Seoul, Korea

* yhlee@yuhs.ac

**Data Availability Statement:** All relevant data are within the paper and its Supporting Information files.

## Abstract

### Background

An orally active synthetic adiponectin receptor agonist, AdipoRon has been suggested to ameliorate insulin resistance, and glucose tolerance. However, the chronic effect of AdipoRon in the vascular dysfunction in type 2 diabetes has not been studied yet. Thus, in this study, we examined whether AdipoRon improves vascular function in type 2 diabetes.

### Methods

Type 2 diabetic (db⁻/db⁻) mice were treated with AdipoRon (10 mg/kg/everyday, by oral gavage) for 2 weeks. Body weight and blood glucose levels were recorded every other day during the experimental period. Diameter of mesenteric arteries was measured. And western blot analysis was performed with mesenteric arteries.

### Results

Pressure-induced myogenic response was significantly increased while endothelium-dependent relaxation was reduced in the mesenteric arteries of db⁻/db⁻ mice. Treatment of AdipoRon normalized potentiated myogenic response, whereas endothelium-dependent relaxation was not affected by treatment of AdipoRon. The expression levels of AdiR1, AdiR2, APPL1, and APPL 2 were increased in the mesenteric arteries of db⁻/db⁻ mice and treatment of AdipoRon did not affect them. Interestingly, AdipoRon treatment increased the phospho-AMPK and decreased MYPT1 phosphorylation in db⁻/db⁻ mice while there was no change in the level of eNOS phosphorylation.

### Conclusion

The treatment of AdipoRon improves vascular function in the mesenteric arteries of db⁻/db⁻ mice through endothelium-independent mechanism. We suggest that MLCP activation through reduced phosphorylation of MYPT1 might be the dominant mechanism in the AdipoRon-induced vascular effect.

**Funding:** This work was funded by a grant from Myoung-Sun Kim Memorial Foundation to SKC (#2017). The funders had no role in study design, data collection and analysis, decision to publish, or preparation of the manuscript.

**Competing interests:** The authors have declared that no competing interests exist.

# Introduction

Obesity is defined as an excessive and abnormal fat accumulation to exert health concerns. Obesity is considered the major factor in the development of various disease such as type 2 diabetes, hypertension, cardiovascular disease, respiratory disease, and osteo-arthritis [1]. Accumulating evidence indicates that obesity frequently occurs with type 2 diabetes and is considered to be a strong risk factor for the development of type 2 diabetes [2,3]. Obesity and type 2 diabetes have deleterious effects on vascular function and create conditions that favor cardiovascular disease [4].

Adiponectin is an important and abundant adipokine secreted from adipocyte and regulates insulin sensitivity and energy homeostasis. The low concentration of adiponectin is associated with various disease such as obesity, diabetes, cardiovascular diseases [5]. Recent studies reported plasma adiponectin level was decreased in the patients with type 2 diabetes, and thiazolidinedione (TZD) administration increased the adiponectin level [6,7]. An experimental study showed that insulin resistance was ameliorated by the replenishment of adiponectin in mice [8]. Thus adiponectin has been focused as potential therapeutic target for the treatment of type 2 diabetes [9]. Adiponectin regulates cellular function via two specific receptors, adiponectin receptor 1 (AdiR1) and adiponectin receptor 2 (AdiR2) [10]. Adaptor protein containing a pleckstrin homology (PH) domain, phosphotyrosine-binding (PTB) domain, and leucine zipper motif 1 (APPL1) is the first identified adapter protein to positively mediate intracellular adiponectin signaling. APPL1 directly binds to the intracellular domain of adiponectin receptor and positively mediates the signaling to the AMP-activated protein kinase (AMPK), p38 mitogen activated protein kinase (MAPK), and peroxisome proliferator-activated receptor α (PPARα) [11]. On the other hand APPL2, an isoform of APPL1, blocks APPL1-mediated insulin-sensitizing effect of adiponectin and thus negatively regulates adiponectin signaling [9].

Recently, an orally active adiponectin receptor agonist, AdipoRon, has been developed and showed similar effects to adiponectin. Like adiponectin, AdipoRon binds to both AdiR1 and AdiR2 at a low molecular concentration and activates AMPK, PPAR, and peroxisome proliferator–activated receptor gamma coactivator 1–alpha (PGC1α) [12]. AdipoRon improved insulin sensitivity and glucose tolerance and lipid metabolism in cultured cells and mice [13]. Furthermore, treatment of AdipoRon improved metabolic function and extended life span in type 2 diabetic mice. Although effects of AdipoRon have been investigated in various pathophysiological states, the effects of AdipoRon on vascular function, specifically in type 2 diabetes have not yet been studied. Therefore, the objectives of the present study were to elucidate whether adiponectin receptor agonist, AdipoRon, improves vascular function in the mesenteric arteries of type 2 diabetic mice and, if so, to determine the mechanisms involved.

# Methods

All experiments were performed according to the Guide for the Care and Use of Laboratory Animals published by US National Institutes of Health (NIH publication No. 85–23, 2011) and were approved by the Ethics Committee and the Institutional Animal Care and Use Committee of Yonsei University, College of Medicine (Approval number: 2017–0173).

## Animal models and tissue preparation

Ten- to 12-week-old male type 2 diabetic mice ($db^-/db^-$) and age-matched heterozygote control mice ($db^-/db^+$) were obtained from Jackson Laboratories. Mice were housed in an AAALAC approved animal facility at Yonsei University. Briefly, mice were housed in plastic cages with stainless steel grid tops at 23~24°C with a 12-hour light/dark cycle and allowed access to commercial rodent chow and water *ad libitum*. Total 30 diabetic mice and 20 control mice

were used in this study. Mice were divided into 4 groups: (1) control mice treated with vehicle for 2 weeks (control mice); (2) control mice treated with AdipoRon (10 mg/kg/everyday, by oral gavage) for 2 weeks; (3) diabetic mice treated with vehicle for 2 weeks (diabetic mice); (4) diabetic mice treated with AdipoRon (10 mg/kg/everyday, by oral gavage) for 2 weeks.

Body weight and blood glucose levels were recorded every other day during the experimental period. At the end of the treatment period, mice were euthanized with isoflurane (5%) followed by the $CO_2$ inhalation. To confirm death, we monitored mice for the several signs such as no rising and falling of chest, no response to toe pinch, no palpable heartbeat, color change opacity in eyes. After we confirm the death, the heart was removed immediately and tissue samples were obtained. To isolate mesenteric artery, the mesenteric small artery beds were removed and placed in ice-cold Krebs-Henseleit (K-H) solution (composition in mmol/L: NaCl, 119; $CaCl_2$, 2.5; $NaHCO_3$, 25; $MgSO_4$, 1.2; $KH_2PO_4$, 1.2; KCl, 4.6; and glucose, 11.1). The third branch of mesenteric arteries (120–150 μm, inner diameter at 40 mmHg) were isolated and cut into 2- to 3-mm segments for subsequent analysis.

## Blood glucose

Blood glucose level was measured in the tail blood samples using a blood glucose meter (Accu-Chek, Roche Diagnostic, Berlin, Germany) in all of the groups of mice after a 6-hour fast.

## Preparation of isolated mesenteric artery

After 2 weeks of treatment, mice were euthanized and mesenteric arteries were isolated and cannulated with glass micropipettes. Krebs-Henseleit (K-H) solution bubbled with a 95% $O_2$ + 5% $CO_2$ gas mixture was perfused into the arteries. The arteries were pressurized to 40 mmHg using pressure-servo control perfusion systems (Living Systems Instruments, St Albans, USA) for 30-minutes equilibration period. A video image analyzer, as described previously [14], monitored the vessel diameter. Intraluminal pressure was increased from 20 to 120 mm Hg in a stepwise manner to measure myogenic response. At the end of the experiments, vessels were superfused with a calcium-free K-H solution containing 1 mM ethylene glycol-bis (2-aminoethylether)-N,N,N′,N′-tetraacetic acid (EGTA) to determine passive diameter. Myogenic response was calculated as the percentage between active and passive diameters. To determine the endothelium-dependent relaxation, pressurized arteries were pre-contracted with thromboxane agonist (U-46619, $10^{-7}$ mol/L), and then cumulative concentrations ($10^{-9}$ to $10^{-5}$ mol/L) of acetylcholine were applied.

## Cell isolation and culture

Vascular smooth muscle cells (VSMCs) were obtained by digestion of freshly isolated aortas of 10-week-old Sprague Dawley rats for 30 minutes in collagenase (1 mg/mL, Worthington Biomedical Corporation, Lakewood Township, NJ, USA) and elastase (0.5 mg/ml, Calbiochem, San Diego, CA, USA) in Dulbecco's Modified Eagle Medium (DMEM, Gibco, Waltham, MS, USA) at 37°C. After trituration and centrifugation, the cells were seeded in culture dishes (Corning, New York, NY, USA) and cultivated in DMEM supplemented with 10% FBS, 100 IU/mL penicillin, and 100 μg/mL streptomycin (Gibco) at 37°C, 5% $CO_2$ with humidified atmosphere. The early passage cells (between 2 and 4) were used.

## Western blot analysis

Mesenteric arteries were isolated and immediately frozen in liquid nitrogen and then homogenized in ice-cold lysis buffer, as described previously [15]. VSMCs were cultured as described

above and stimulated with vehicle ($H_2O$) or phenylephrine (PE, 5 μM) or PE (5 μM) + AdipoRon (50 μM) for 10 minutes at 37˚C. Cells were then lysed in lysis buffer for 10 minutes on ice. Soluble extracts were prepared by centrifugation with 10,000 *g* for 10 minutes at 4˚C. Western blot analysis was performed for AdiR1 (1:1000 dilution; Abcam #ab70362, Cambridge, MA, USA), AdiR2 (1:1000 dilution; Abcam #ab77613, Cambridge, MA, USA), APPL1 (1:1000 dilution; Santa Cruz Biotechnology #SC-271901, Santa Cruz, CA, USA), APPL2 (1:1000 dilution; Santa Cruz Biotechnology #SC-271084, Santa Cruz, CA, USA), phosphorylated AMPK (1:1000 dilution; Abcam #2532s, Cambridge, MA, USA), total AMPK (1:1000 dilution; Abcam #2535s, Cambridge, MA, USA), phosphorylated eNOS (1:1000 dilution; Cell signaling #9571, Danvers, MA, USA), total eNOS (1:1000 dilution; Cell signaling #9586s, Danvers, MA, USA), and phosphorylated MYPT1 (myosin phosphatase target subunit 1, 1:500 dilution; Bioss #bs-3288R, Woburn, MA, USA), and total MYPT1 (1:500 dilution; Bioss #bs-3787R, Woburn, MA, USA). Blots were stripped and then reprobed with the β-actin antibody (1:2000 dilution; Abcam #3280, Cambridge, MA, USA) to verify the equal loading between the samples.

## Drugs

The following drugs were used: U-46619 (Tocris Bioscience, Ellisville, MO, USA); acetylcholine (Sigma-Aldrich, St Louis, MO, USA); AdipoRon (Cayman Chemicals, Ann Arbor, MI, USA); PE (Sigma-Aldrich); the general laboratory reagents (Sigma-Aldrich)

## Data and statistical analysis

Results are expressed as mean ± SD. Differences between values were evaluated using a one or two-way repeated-measures analysis of variance (ANOVA) followed by Bonferroni *post-hoc* test. Values of $P < 0.05$ were statistically significant. For all experiments measuring diameter, the n-values mean number of vessels derived from each different animals. Accordingly, n-values also mean number of animals.

# Results

## Effect of AdipoRon on blood glucose level and body weight

Body weight (Fig 1A) and blood glucose levels (Fig 1B) were significantly higher in diabetic mice (52.0 ± 2.5 g, 531.5 ± 28.7 mg/dl, respectively) compared to control mice (25.3 ± 0.4 g, 132.6 ± 9.4 mg/dl, respectively). Treatment of AdipoRon did not affect the body weight and blood glucose level in both diabetic mice (55.52 ± 1.06 g, 576.8 ± 13.1 mg/dl, respectively) and control mice (26.3 ± 1.20 g, 122.4 ± 5.2 mg/dl, respectively).

## Effect of AdipoRon on myogenic response and endothelium-dependent relaxation in mesenteric arteries

To determine the effect of AdipoRon on the vascular reactivity, we evaluated myogenic response and endothelium-dependent relaxation in isolated mesenteric arteries of all groups of mice. Myogenic response was significantly increased in mesenteric arteries of diabetic mice compared to control mice. Treatment of AdipoRon significantly reduced myogenic response in the type 2 diabetic mice (Fig 2A). Endothelium-dependent relaxation was significantly decreased in mesenteric arteries of diabetic mice compared to control mice. Treatment of AdipoRon did not affect the endothelium-dependent relaxation in the all groups of the mice. (Fig 2B).

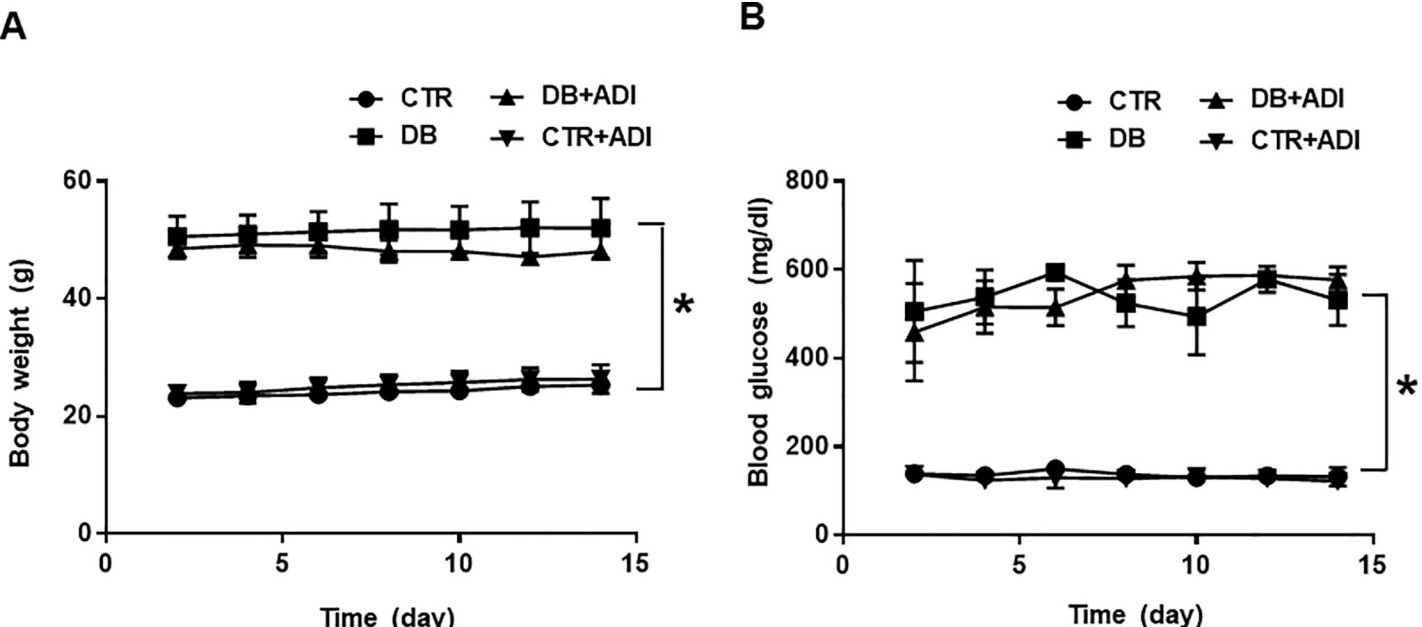

**Fig 1. Effects of AdipoRon on body weight and blood glucose.** CTR: control mice treated with vehicle; CTR+ADI (AdipoRon): control mice treated with AdipoRon; DB: diabetic mice treated with vehicle; DB+ADI (AdipoRon): diabetic mice treated with AdipoRon. (A) Comparison of body weight between groups (n = 5), (B) Comparison of blood glucose concentrations between groups (n = 5) * P < 0.05.

### Effect of direct administration of AdipoRon in the mesenteric arteries

To determine the direct effect of AdipoRon on the vascular reactivity, we administered AdipoRon in the pre-constricted mesenteric arteries. Mesenteric arteries were pressurized at 40 mmHg and pre-contracted with U-46619 ($10^{-7}$ mol/L), and then cumulative concentrations ($5 \times 10^{-6}$ to $2 \times 10^{-4}$ mol/L) of AdipoRon were applied. AdipoRon induced dose-dependent relaxation in both control mice and diabetic mice (Fig 2C). There was no significant difference in AdipoRon-induced relaxation between control mice and diabetic mice.

### Effects of AdipoRon on expression levels of AdiR1, AdiR2, APPL1, and APPL2

To determine whether treatment of AdipoRon affects adiponectin signaling molecules, we measured the expression levels of adiponectin receptors and signaling molecules in mesenteric arteries. We found that expression levels of adiponectin receptors, AdiR1 and AdiR2 were significantly increased in the mesenteric arteries of diabetic mice compared to control mice (Fig 3A&3B). Treatment of AdipoRon did not affect the elevated expression levels of AdiR1 and AdiR2. We also measured expression levels of downstream signaling molecules of adiponectin receptor, APPL1 and APPL2, in the mesenteric arteries. The expression levels of APPL1 and APPL2 were increased in the mesenteric arteries of diabetic mice (Fig 3C&3D). The elevated levels of APPL1 and APPL2 were not changed by treatment of AdipoRon.

### Effects of AdipoRon on changes in AMPK and eNOS

We evaluated expression and phosphorylation levels of AMPK, a downstream signaling molecule of adiponectin receptor, in the mesenteric arteries. Phosphorylation level of AMPK was significantly reduced in the mesenteric arteries of diabetic mice compared to control mice. Interestingly, treatment of AdipoRon almost normalized the phosphorylation level of AMPK

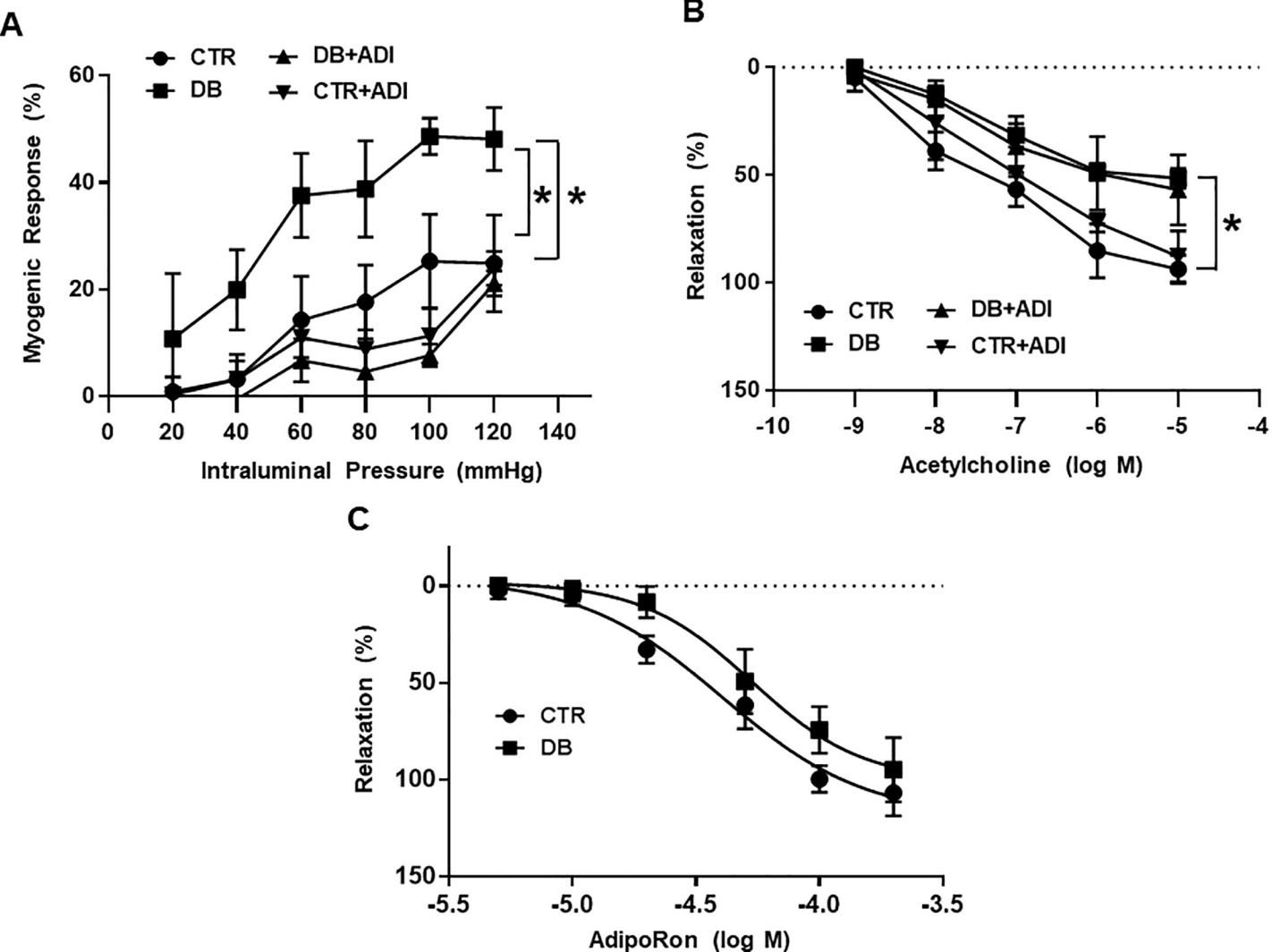

**Fig 2. Effects of AdipoRon on myogenic response and endothelium-dependent relaxation in mesenteric arteries.** CTR: control mice treated with vehicle; CTR+ADI (AdipoRon): control mice treated with AdipoRon; DB: diabetic mice treated with vehicle; DB+ADI (AdipoRon): diabetic mice treated with AdipoRon. (A) Summarized data for the effects of AdipoRon on pressure-induced myogenic response. Changes in inner diameter were measured in response to 20 mmHg stepwise increases in intraluminal pressure in the K-H solution (active diameter) or $Ca^{2+}$ free K-H solution (passive diameter). (n = 5). (B) Summarized data for endothelium-dependent relaxation in response to cumulative doses of acetylcholine ($10^{-9}$ to $10$-$5^{-5}$ mol/L) in the mesenteric arteries pre-contracted with U-46619 ($10^{-7}$ mol/L). (n = 5). (C) Dose-response ($5 \times 10^{-6}$ to $2 \times 10^{-4}$ mol/L) curves for AdipoRon in the mesenteric arteries pre-contracted with U-46619 ($10^{-7}$ mol/L) * P < 0.05.

in diabetic mice (Fig 4A). We also measured the phosphorylation of eNOS in the mesenteric arteries to examine whether eNOS expression and/or phosphorylation was affected by AdipoRon treatment. Phosphorylation level of eNOS was significantly decreased in diabetic mice compared to control mice. Treatment of AdipoRon did not affect eNOS phosphorylation level in diabetic mice (Fig 4B).

## Effects of AdipoRon on changes in MYPT1 in the mesenteric arteries and VSMCs

To determine whether normalization of myogenic responses in the AdipoRon-treated diabetic mice were resulted from regulation of myosin light chain phosphatase (MLCP) activity, we

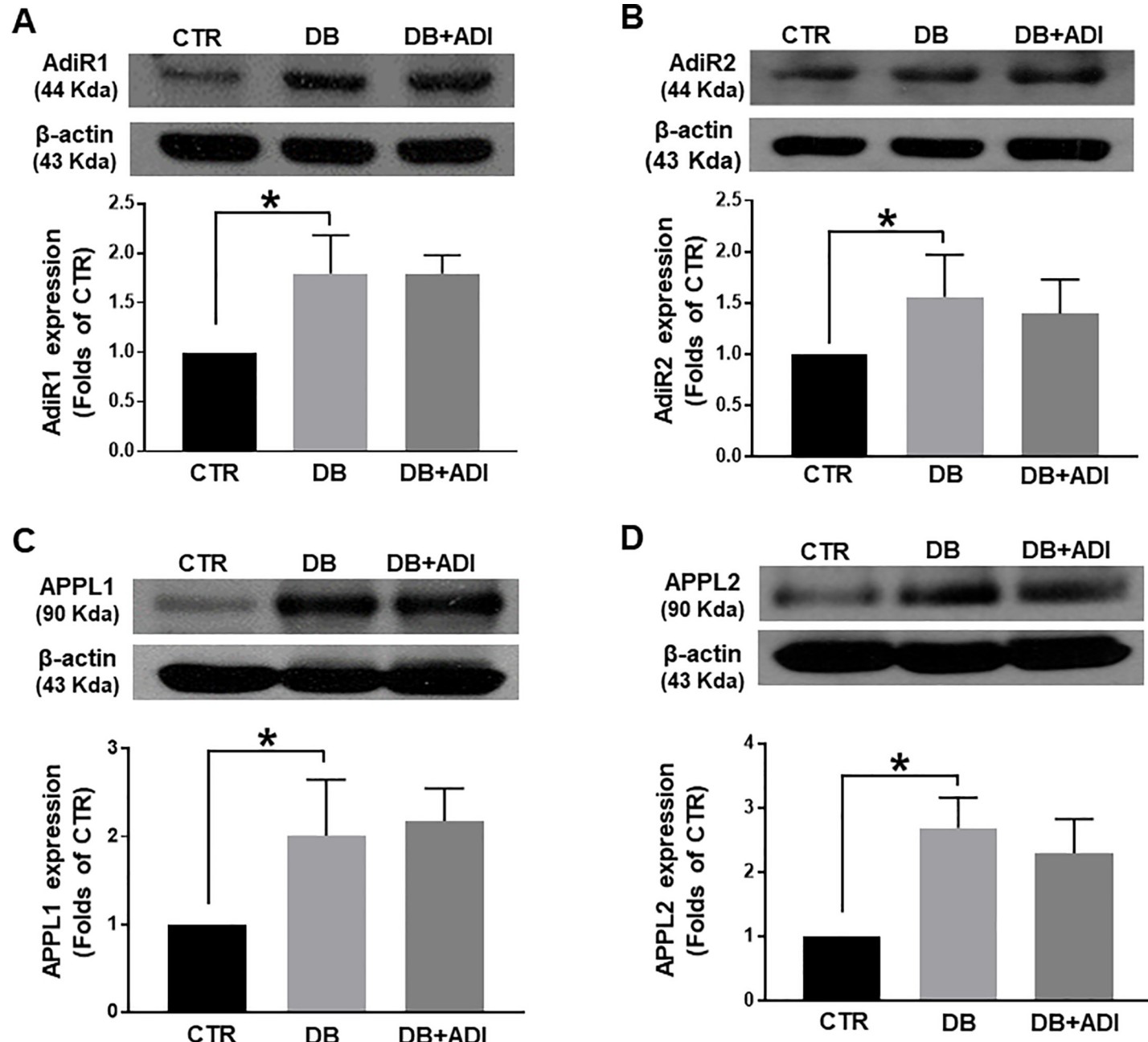

**Fig 3. Effects of AdipoRon on changes in adiponectin signaling molecules, AdiR1, AdiR2, APPL1, and APPL2 in mesenteric arteries.** CTR: control mice treated with vehicle, DB: diabetic mice treated with vehicle, and DB+ADI (AdipoRon): diabetic mice treated with AdipoRon. Representative western blot analysis and quantitative data for AdiR1 (A), AdiR2 (B), APPL1 (C), and APPL2 (D), (n = 4). Each group of blots was cropped from a same gel and membrane. * P < 0.05.

quantified the phosphorylation level of MYPT1 of MLCP in the mesenteric arteries. Phosphorylation level of MYPT1 at Thr[850] was significantly increased in the mesenteric arteries of diabetic mice compared to control mice. Treatment of AdipoRon reduced the elevated MYPT1 phosphorylation at Thr[850] level in diabetic mice (Fig 5A). To confirm the effect of AdipoRon on MYPT1 phosphorylation, we performed western blot analysis with primary cultured rat aortic VSMCs. To determine the level of phosphorylated MYPT1 at Thr[850], we stimulated VSMCs with PE (5 μM) since it t has been well known that PE significantly increases MYPT1

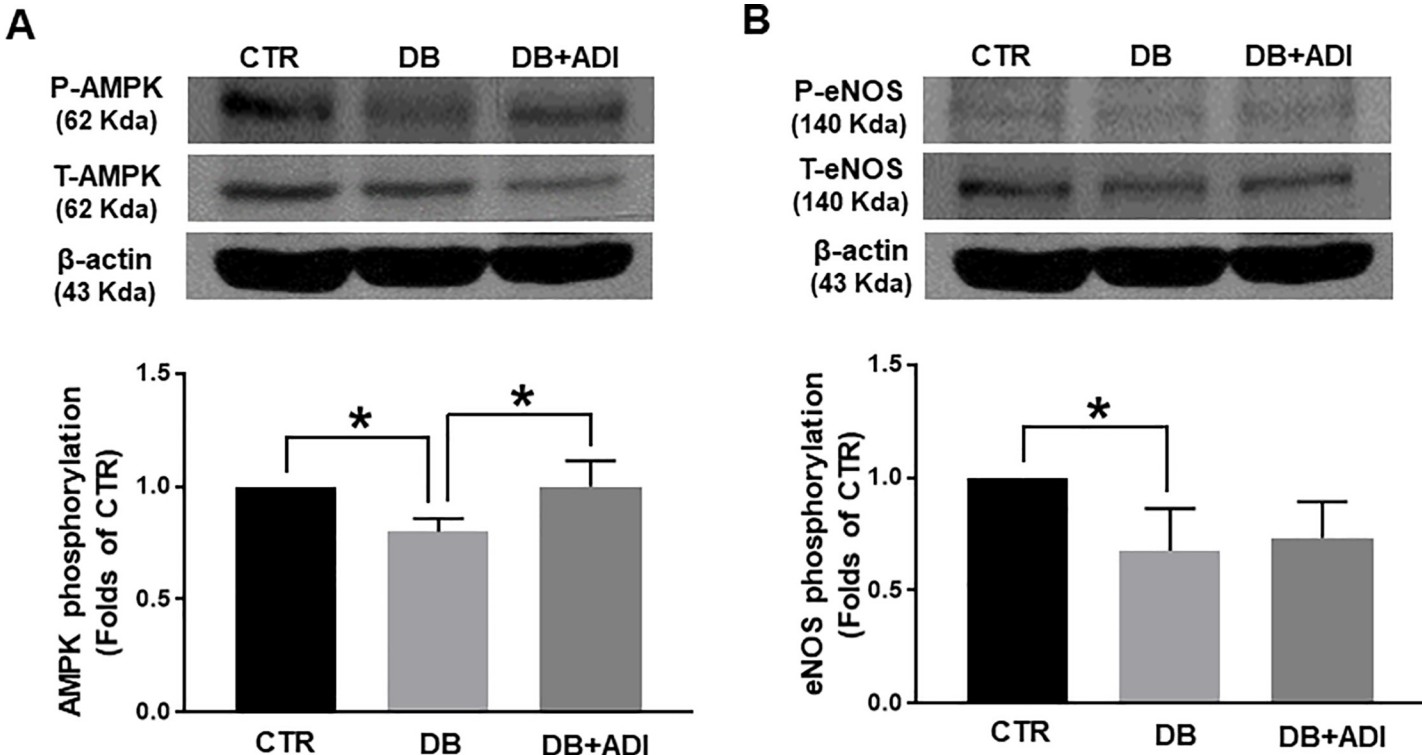

**Fig 4. Effects of AdipoRon on changes in on changes in AMPK and eNOS.** CTR: control mice treated with vehicle, DB: diabetic mice treated with vehicle, and DB +ADI (AdipoRon): diabetic mice treated with AdipoRon. Representative western blot analysis and quantitative data for AMPK (A) and eNOS (B), (n = 4). The blots of AMPK (A) and eNOS (B) were cropped from a same gel and membrane. * P < 0.05.

phosphorylation at Thr$^{850}$ [16]. PE stimulation increased phosphorylated MYPT1 at Thr$^{850}$ whereas co-stimulation of PE and AdipoRon did not increase MYPT1 phosphorylation in rat aortic VSMCs (Fig 5B).

## Discussion

The main findings from this study are (1) myogenic response and endothelium-dependent relaxation were impaired in mesenteric arteries of type 2 diabetic mice; (2) treatment of AdipoRon normalized myogenic response and did not affect the endothelium-dependent relaxation; (3) treatment of AdipoRon did not affect to the expression levels of AdiR1, AdiR2, APPL1, APPL2, and phosphorylation level of eNOS; (4) treatment of AdipoRon increased the phosphorylation level of AMPK and decreased MYPT1 phosphorylation in the mesenteric arteries of type 2 diabetic mice.

Adiponectin is a novel cytokine secreted from adipose tissue that exerts various actions such as insulin-sensitizing, glucose-lowering, and anti-inflammatory effects [17]. In the last years it has been reported that adiponectin also has vasoprotective effect and low level of adiponectin is associated with various cardiovascular complications [18]. However, many questions regarding the effect of adiponectin on vascular function remain unclear. In the present study, we used novel and orally active adiponectin receptor agonist, AdipoRon, to study vascular effect of adiponectin signaling in the mesenteric arteries of type 2 diabetic mice.

AdipoRon has been recently developed and reported to mimic effects of adiponectin. Previous study showed that AdipoRon ameliorated insulin resistance and extended lifespan of type 2 diabetic mice [12]. However, in the present study, we did not observe the glucose lowering

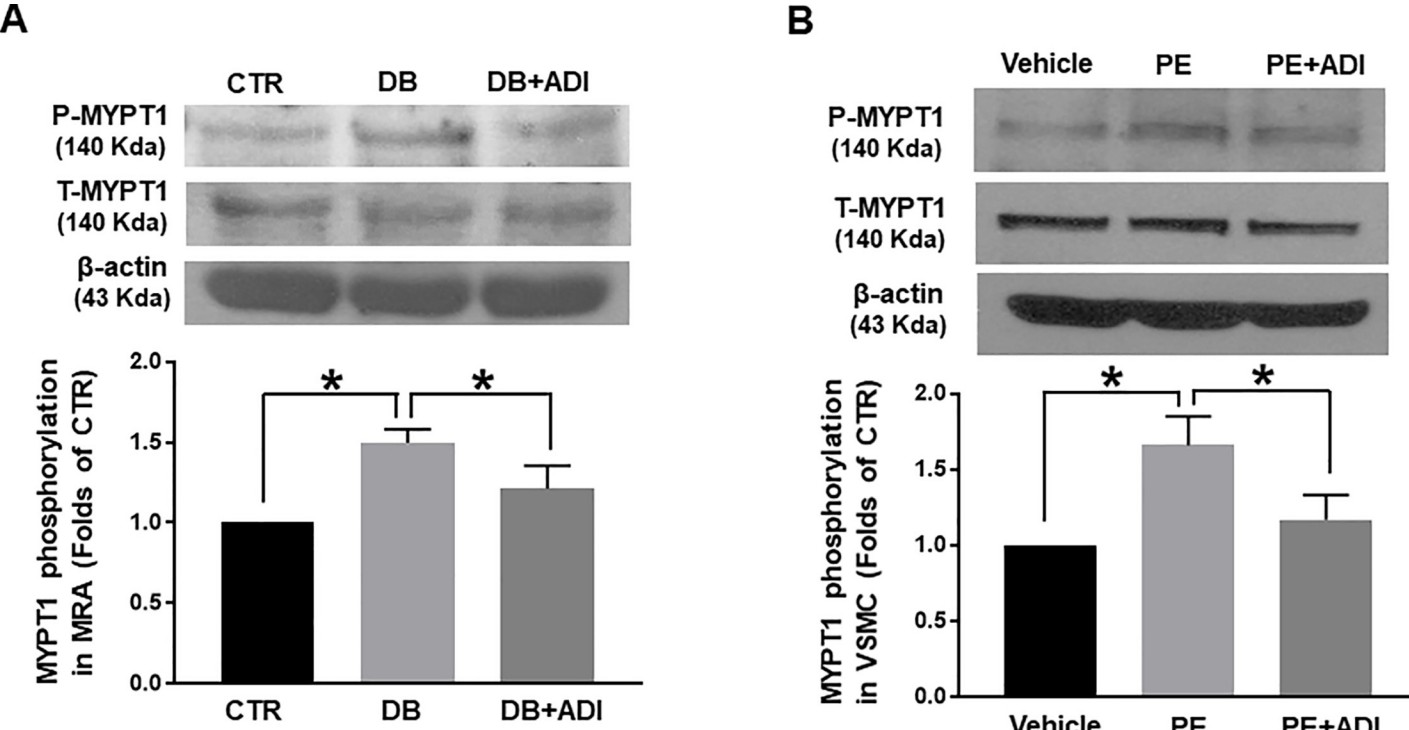

**Fig 5. Effect of AdipoRon on changes in MYPT1 at Thr$^{850}$ in mesenteric arteries and rat aortic vascular smooth muscle cells (VSMCs).** (A) CTR: control mice treated with vehicle, DB: diabetic mice treated with vehicle, and DB+ADI (AdipoRon): diabetic mice treated with AdipoRon. Representative western blot analysis and quantitative data for MYPT1 in mesenteric arteires (n = 4). (B) Vehicle: VSMCs stimulated with vehicle, PE: VSMCs stimulated with phenylephrine (5 μM), PE+ADI: VSMCs co-stimulated with PE (5 μM) and AdipoRon (ADI, 50 μM). Representative western blot analysis and quantitative data for MYPT1 in VSMCs (n = 4). Each group of blots was cropped from a same gel and membrane. $^*$ P < 0.05.

effect of the AdipoRon in the same animal model. There could be two possibilities to describe about this discrepancy. The first one is that the dose of AdipoRon we used was not sufficient to affect to glucose homeostasis but it could be the specific dose to improve the vascular functionality. The second possibility is that AdipoRon may have different effect between the long-term treatment and acute treatment. In the previous study, only acute effect of AdipoRon was observed in the type 2 diabetic mice [12]. Further studies are needed to elucidate the mechanisms involved in glucose lowering effect of AdipoRon.

Vascular functionality could be defined by the myogenic response and endothelium-dependent relaxation. The myogenic response is an intrinsic response of arteries that plays a critical role in the local regulation of blood flow and characterized by vasoconstriction in response to an increase of intraluminal pressure and vasodilatation in response to a decrease in intraluminal pressure [19]. It has been suggested that impaired small artery function is attributable to the enhanced myogenic response in diabetic animals [20]. In the present study, we observed potentiated myogenic response in the mesenteric arteries of type 2 diabetic mice (Fig 2A). Interestingly, treatment of AdipoRon normalized potentiated myogenic response. However, impaired endothelium-dependent relaxation was not affected by the treatment of AdipoRon (Fig 2B). Furthermore, we did not find significant difference in the dose-response relationship of AdipoRon between control mice and diabetic mice (Fig 2C). We assume that AdipoRon itself could induce the relaxation in the vessels but it does not improve endothelial function *in vivo*. These data are in accordance with previous studies that AdipoRon induced vasorelaxation though endothelium-independent mechanism [21].

In order to define the underlying mechanism, we measured the expression levels of adiponectin receptor signaling molecules. Adiponectin exert its effect through two receptors, AdiR1 and AdiR2 [22,23]. It has been confirmed AdipoRon bound both AdiR1 and AdiR2 [12]. We observed that expression levels of both AdiR1 and AdiR2 are increased in the mesenteric arteries of diabetic mice compared to control mice (Fig 3). These results are consistent with the previous study showed AdiR1 and AdiR2 were increased in the retina of diabetic human and mice [24]. On the other hand, it has been reported that abundance of AdiR1 and AdiR2 was not different between healthy control and type 2 diabetic patients [25]. Despite of this discrepancy, we assume that the elevation of AdiR1 and AdiR2 might represent a mechanism of compensation in the mesenteric arteries of diabetic mice since this is the first report to compare expression levels of AdiR1 and AdiR2 in the vascular tissues.

Since the specific antibodies to detect the activated AdiR1 and AdiR2 were not available, we observed the expression levels of APPL1 and APPL2, the downstream effectors for adiponectin. Recent studies have been suggested that APPL1 positively mediate adiponectin signaling in various cell types such as endothelial cells and skeletal muscle cells [26]. In contrast, APPL2 seems to negatively regulate adiponectin signaling by competing with APPL1 [9]. We found that expression levels of both APPL1 and APPL2 were increased in the mesenteric arteries of diabetic mice compared to control mice. Although expression level of APPL1 was increased in diabetic mice, APPL2 expression was also increased, which results in the reduced action of APPL1. Thus the adiponectin signaling seems to be altered in the mesenteric arteries of diabetic mice. AdipoRon did not affect to this alteration of adiponectin receptor signaling in diabetic mice.

As it has been reported that adiponectin stimulates the phosphorylation of eNOS through AMPK activation [27], we examined whether AdipoRon could affect phosphorylation levels of AMPK and eNOS. We found that phosphorylated AMPK was decreased in the mesenteric arteries of diabetic mice compared to control mice and treatment of AdipoRon increased phosphorylated AMPK (Fig 4A). However, we did not observe any change in eNOS phosphorylation level (Fig 4B). These data were consistent with previous study showed that AdipoRon exerted renoprotective effect by increasing of phosphorylated AMPK in the type 2 diabetic mice [28]. On the other hand, our data are not in accordance with the study showed that AdipoRon ameliorated diabetic nephropathy by increasing eNOS phosphorylation [29]. AdipoRon has been recently developed, thus there are not many studies to demonstrate the signaling mechanism regarding the eNOS. Further studies are needed to explore exact mechanism.

To define how AdipoRon affects vascular responses, we observed the phosphorylation level of MYPT1. It has been well known that phosphorylation of MYPT1 at Thr[850] results in inhibition of MLCP activity [30]. We found that MYPT1 phosphorylation at Thr[850] was increased in the mesenteric arteries of diabetic mice compared to control mice. Treatment of AdipoRon significantly decreased the phosphorylation level of MYPT1 (Fig 5). To confirm the effect of AdipoRon on MYPT1 phosphorylation, we observed MYPT1 phosphorylation in primary cultured rat aortic VSMCs. We found that PE stimulation significantly increased MYPT1 phosphorylation at Thr[850] while co-stimulation of PE and AdipoRon did not affect the MYPT1 phopshorylation in VSMCs. From these results, we assume that AdipoRon could affect $Ca^{2+}$ sensitization mechanism. These data are in consistent with previous finding showed that AdipoRon induced vasorelaxation while $Ca^{2+}$ concentration was not changed in the pressurized rat cremaster arteries [21]. Further studies are needed to define the mechanism of action of AdipoRon in the vasculature. Despite the limitations of the study, this is the first study to demonstrate a protective role of AdipoRon in vascular dysfunction of type 2 diabetes.

## Conclusion

Pressure-induced myogenic response and endothelium-dependent relaxation are impaired in mesenteric arteries of diabetic mice. Treatment of AdipoRon normalized potentiated myogenic response, whereas endothelium-dependent relaxation was not affected by AdipoRon. AdipoRon treatment increased the phospho-AMPK and decreased MYPT1 phosphorylation in diabetic mice while there was no change in the level of eNOS phosphorylation. MLCP activation through reduced phosphorylation of MYPT1 might be the dominant mechanism in the AdipoRon-induced vascular effect. In conclusion, we suggest that the novel adiponectin receptor agonist, AdipoRon, as a vasoprotective agent to ameliorate the vascular dysfunction in type 2 diabetes.

## Supporting information

**S1 Fig.**
(PDF)

## Author Contributions

**Conceptualization:** Soo-Kyoung Choi, Young-Ho Lee.

**Data curation:** Soo-Kyoung Choi.

**Funding acquisition:** Soo-Kyoung Choi, Seonhee Byeon.

**Investigation:** Soo-Kyoung Choi, Youngin Kwon, Seonhee Byeon, Chae Eun Haam.

**Supervision:** Young-Ho Lee.

**Writing – original draft:** Soo-Kyoung Choi.

**Writing – review & editing:** Soo-Kyoung Choi.

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
