## [Decision Letter · Decision Letter 0]

14 Nov 2019

PONE-D-19-29078

AdipoRon, adiponectin receptor agonist, improves vascular function in the mesenteric arteries of type 2 diabetic mice

PLOS ONE

Dear Dr. Choi,

Thank you for submitting your manuscript to PLOS ONE. After careful consideration, we feel that it has merit but does not fully meet PLOS ONE’s publication criteria as it currently stands. Therefore, we invite you to submit a revised version of the manuscript that addresses the points raised during the review process. The lack of novelty mentioned by reviewer 2 is no criterion for publication in PLOS ONE. However you should provide evidence about the mechanisms by which AdipoRon activates phosphorylation of AMPK and decreases MYPT phosphorylation as suggested by reviewer 1.

We would appreciate receiving your revised manuscript by Dec 29 2019 11:59PM. To enhance the reproducibility of your results, we recommend that if applicable you deposit your laboratory protocols in protocols.io, where a protocol can be assigned its own identifier (DOI) such that it can be cited independently in the future. For instructions see: http://journals.plos.org/plosone/s/submission-guidelines#loc-laboratory-protocols

We look forward to receiving your revised manuscript.

Kind regards,

Michael Bader

Academic Editor

PLOS ONE

Journal Requirements:

1. PLOS ONE now requires that authors provide the original uncropped and unadjusted images underlying all blot or gel results reported in a submission’s figures or Supporting Information files. This policy and the journal’s other requirements for blot/gel reporting and figure preparation are described in detail at https://journals.plos.org/plosone/s/figures#loc-blot-and-gel-reporting-requirements and https://journals.plos.org/plosone/s/figures#loc-preparing-figures-from-image-files. When you submit your revised manuscript, please ensure that your figures adhere fully to these guidelines and provide the original underlying images for all blot or gel data reported in your submission. See the following link for instructions on providing the original image data: https://journals.plos.org/plosone/s/figures#loc-original-images-for-blots-and-gels.

Reviewers' comments:

Reviewer's Responses to Questions

**Comments to the Author**

1. Is the manuscript technically sound, and do the data support the conclusions?

Reviewer #1: Yes

Reviewer #2: Partly

2. Has the statistical analysis been performed appropriately and rigorously? 

Reviewer #1: Yes

Reviewer #2: Yes

3. Have the authors made all data underlying the findings in their manuscript fully available?

Reviewer #1: Yes

Reviewer #2: Yes

4. Is the manuscript presented in an intelligible fashion and written in standard English?

Reviewer #1: Yes

Reviewer #2: Yes

5. Review Comments to the Author

Reviewer #1: The author Dr Choi and the colleagues demonstrated that AdipoRon improves vascular function in the mesenteric arteries in diabetic mice model via an endothelium independent manner and activated MLCP by MYPT1 could be the reason for this induced effect. Although multiple observations and experiments were made, it is lack of convincing evidence to support the current conclusion, robust mechanism is needed as well.

1. AdipoRon is an adiponectin receptor activator and it has been proved that mesenteric arteries expressed AdiRs. The unchanged the protein levels does not equal to unchanged activity. Nitric oxide blocker or adiponectin receptor specific knockout model need to be applied to solidate the finding.

2. The evidence to support the finding that demonstrate adipoRon can directly activate MYPT of MLCP in the mesenteric arteries need to further obtain evidence from cultured cells. Meanwhile, why is the MYPT is the signals AdipoRon relied on? Cause-effect approaches is required to enforce the current conclusion.

Reviewer #2: The authors of the present study have examined the vasorelaxant effects of AdipoRon, an adiponectin receptor agonist, using mesenteric arteries from type 2 diabetic mice. Although the observed findings appear to be interesting, there are concerns regarding the novelty of the present study as noted below.

Introduction:

i) The Introduction section is very weak. The first two paragraphs are descriptive, and they appear to be a review of the previously known facts about obesity, diabetes, and adiponectin receptor signaling.

ii) The very first article on AdipoRon, an adiponectin receptor agonist, was published in 2013. In the third paragraph of Introduction, the authors have cited only this article by Okada-Iwabu (Nature 2013).

iii) From the Introduction section, it is clear that the present manuscript lacks novelty. Previously, it has been reported that AdipoRon induces vasorelaxation by mechanisms independent of endothelium-dependent relaxing factors and AMPK activation (Hong et al. Microcirculation 23, 207-220, 2016).

Methods and Results:

i) Using pressure and wire myography, Hong et al have previously reported the vasorelaxant effects of AdipoRon in different vascular beds, including skeletal muscle, cerebral, coronary and mesenteric arteries. Thus, the findings of the present study are not novel (Microcirculation 23, 207-220, 2016).

ii) For the present study using type 2 diabetic db−/db− mice, it is unclear as to why the authors have used db−/db+ mice as the control group instead of the wild-type mice.

6. PLOS authors have the option to publish the peer review history of their article (what does this mean?). If published, this will include your full peer review and any attached files.

Reviewer #1: No

Reviewer #2: No

---

## [Author Response · Author response to Decision Letter 0]

28 Jan 2020

We thank the Editor and Reviewers for their careful review and do appreciate the constructive comments that strengthen our manuscript. The manuscript was revised in accordance with the suggestions of the reviewers. A rebuttal letter was uploaded as a separate file.

---

## [Decision Letter · Decision Letter 1]

26 Feb 2020

AdipoRon, adiponectin receptor agonist, improves vascular function in the mesenteric arteries of type 2 diabetic mice

PONE-D-19-29078R1

Dear Dr. Choi,

We are pleased to inform you that your manuscript has been judged scientifically suitable for publication and will be formally accepted for publication once it complies with all outstanding technical requirements. If you have data to address the comments of reviewer 1, please add them, but this is not essential for the acceptance of the manuscript.

With kind regards,

Michael Bader

Academic Editor

PLOS ONE

Additional Editor Comments (optional):

Reviewers' comments:

Reviewer's Responses to Questions

**Comments to the Author**

1. If the authors have adequately addressed your comments raised in a previous round of review and you feel that this manuscript is now acceptable for publication, you may indicate that here to bypass the “Comments to the Author” section, enter your conflict of interest statement in the “Confidential to Editor” section, and submit your "Accept" recommendation.

Reviewer #1: (No Response)

2. Is the manuscript technically sound, and do the data support the conclusions?

Reviewer #1: Partly

3. Has the statistical analysis been performed appropriately and rigorously? 

Reviewer #1: Yes

4. Have the authors made all data underlying the findings in their manuscript fully available?

Reviewer #1: Yes

5. Is the manuscript presented in an intelligible fashion and written in standard English?

Reviewer #1: Yes

6. Review Comments to the Author

Reviewer #1: All the comments are not fully addressed. In addition, APPL1 and APPL2's cellular location must be determined. Cause-effect approach need to be determined to prove AdipoRon regulates MYPT to exhibit its effect.

7. PLOS authors have the option to publish the peer review history of their article (what does this mean?). If published, this will include your full peer review and any attached files.

Reviewer #1: No

---

## [Editor Report · Acceptance letter]

5 Mar 2020

PONE-D-19-29078R1 

AdipoRon, adiponectin receptor agonist, improves vascular function in the mesenteric arteries of type 2 diabetic mice 

Dear Dr. Choi:

I am pleased to inform you that your manuscript has been deemed suitable for publication in PLOS ONE. Congratulations! Your manuscript is now with our production department. 

With kind regards,

on behalf of

Prof. Michael Bader 

Academic Editor

PLOS ONE